# Spatiotemporal Variations in Water Quality of the Transboundary Shari-Goyain River, Bangladesh

Debasish Pandit [1,2], Mohammad Mahfujul Haque [3], Ahmed Harun-Al-Rashid [1], Bishwajit Sarker [4], Mohammad Amzad Hossain [5], Petra Schneider [6,*] and Mrityunjoy Kunda [1,*]

1 Department of Aquatic Resource Management, Sylhet Agricultural University, Sylhet 3100, Bangladesh; dpandit.sau@gmail.com (D.P.); rashidmha.arm@sau.ac.bd (A.H.-A.-R.)
2 Department of Oceanography, Khulna Agricultural University, Khulna 9100, Bangladesh
3 Department of Aquaculture, Bangladesh Agricultural University, Mymensingh 2202, Bangladesh; mmhaque.aq@bau.edu.bd
4 Department of Agricultural Statistics, Sylhet Agricultural University, Sylhet 3100, Bangladesh; bishwajit.stat@sau.ac.bd
5 Department of Fish Biology and Genetics, Sylhet Agricultural University, Sylhet 3100, Bangladesh; mamzad.fbg@sau.ac.bd
6 Department for Water, Environment, Civil Engineering and Safety, University of Applied Sciences, Magdeburg-Stendal, Breitscheidstraße 2, D-39114 Magdeburg, Germany
* Correspondence: petra.schneider@h2.de (P.S.); kunda.arm@sau.ac.bd (M.K.)

**Abstract:** This study aimed to investigate the seasonal and spatial variations in water quality parameters and determine the main contamination sources in the Shari-Goyain River, Bangladesh. Therefore, surface water was sampled monthly from six sampling sites, where six water quality parameters were evaluated. Data were analyzed by applying the Canadian Council of Ministers of the Environment (CCME) water quality index (WQI) and multivariate statistical methods. The results reveals that most of the examined water quality parameters crossed the acceptable range, and significant variations were observed spatiotemporally ($p < 0.05$). Based on the CCME WQI value, the water quality of the river is classified as poor to marginal with a score range between 33.40 and 51.30. This range of values demonstrates that the river's water quality is far from desirable for aquatic life and that it is being impacted and deteriorated by external drivers. Principal component analysis (PCA) retained two principal components (Factors 1 and 2), explaining about 79.17% of the total variance in the studied parameters and identified acidic pollution sources. Cluster analysis also reveals relative differences in water quality throughout sites and seasons, which supported the CCME WQI and PCA. Finally, Kruskal-Wallis one-way analysis of variance by ranks has identified coal mine drainage (CMD) as the main pollutant source for the Shari-Goyain River. In order to mitigate the CMD impact on land and water, different nature-based solutions are proposed, particularly passive mine water treatment approaches through constructed wetlands that could also mitigate the transboundary waters problem.

**Keywords:** riverine water; water quality; coal mine drainage; multivariate statistics

## 1. Introduction

Freshwater wetlands like rivers, ponds, lakes, etc. are the main inland water resources that meet the day-to-day demand for water supply for industrial, agricultural, and domestic purposes in many countries [1,2]. However, these limited resources of waters are in danger due to pollution, mostly caused by several manmade factors. Mining operations, industrial manufacturing, power generation, agricultural activities, and other manmade factors contribute to the pollution of water resources which ultimately affects human civilization [3–5]. Coal mining is the biggest challenge because of its involvement in local, regional, and global water pollution [6]. Underground and open-cut mining operations are

the widely used methods for coal mining that cause noticeable environmental problems that alter the physicochemical and biological variables of the mining areas and the surrounding environment [7]. Discharges of polluted water from mining activities deteriorate water quality through mixing with not only surface water but also with the groundwater table [8–10]; thus they have a far-reaching impacts on human society and ecological units. Acid mine drainage is mostly responsible for the decline in biodiversity and deterioration of water quality in the rivers at coal mining regions which make the water loaded with harmful trace metals [11–13]. Moreover, acid mine drainage leads to decreased pH and redox conditions that mobilize the bound heavy and trace metals from soils and rivers [14]. The noxious coal mine effluents damage the flora and fauna of the ecosystem and thus have significant negative impacts on the primary productivity, water quality, and diversity of fish in the river riverine fishes [15]. Notably, coal mine drainage (CMD) from upstream via several transboundary rivers including the Jadukata, the Someshwari, and the Shari-Goyain contributes to water pollution in the *haor* basin of northeastern Bangladesh [16,17].

There are several instances of massive fish kills in the Shari-Goyain River due to coal mining activities [18]. Its upstream segment in India is known as the Myntdu River, which originates at Mihmyntdu [19]. It enters Bangladesh through Jaintiapur upazila (sub-district) in Sylhet district and flows through Jaintiapur, Gowainghat, Sylhet Sadar, and Chhatak upazilas of Sylhet district. It meets the Surma River near Chhatak upazila of Sunamganj district [17,20]. Similar to other rivers in Bangladesh, it contributes significantly to the local population's economy by providing domestic water, creating employment, reducing poverty, providing animal protein, etc. [21,22]. Moreover, several families actively participate in fishing, ensuring a year-round source of income [17,23]. It is a flashy hilly river, and thus transports a huge amount of sediments, coarse sands, and boulders from upstream [20,24]. This river sometimes also transports a large amount of abandoned coals with the drainage coming from the upstream catchment areas of Meghalaya in India [18]. Thousands of people depend on sand mining [20,25] and coal dust collection from this river bed for their livelihoods [18]. Several significant mining projects are now underway in Meghalaya, including strip mines for coal extraction and open-pit, hard rock mines. These coal mining activities are done without environmental consideration thus adversely affect the water resources of the Jaintia Hills district of Meghalaya [19]. The polluted acid mine drainage finds its way into the Myntdu River and flows downstream to the Shari-Goyain River. The toxic pollutants also contaminates the drinking water and fish, leading to adverse health effects such as mental disorders, weakness, headaches, abdominal cramps, diarrhea, and anemia in both the cattle and human. As a result, the medical costs have increased, leading to a decline in the socio-economic status of the riverbank dwellers who rely on the river's resources. Thus, upstream polluted water possibly has a potential impact on the fisheries' resources of vast area of aquatic habitat in the Shari-Goyain River [17,26]. In this setting, the issue becomes a transboundary resource problem with an upstream–downstream dimension. Altogether, 54 transboundary rivers pass through Bangladesh and India, all of which are components of the Ganga-Brahmaputra-Meghna (basin's drainage system. Bangladesh's capacity to ensure the availability of food and water relies heavily on the Padma, the Jamuna, and the Meghna Rivers and their associated streams [27–29]. India as the upstream country has its prospect to use its geographic position to gain an advantage in exploiting the water resources at its upper riparian areas. However, the absence of transparent data and such worries over transboundary rivers may lead to a more serious dispute between two normally cordial neighbours [29]. With a population of around 169 million, Bangladesh is heavily dependent on its fisheries resources for food and livelihoods. India, with a population of 1.3 billion, is also increasingly dependent on its fisheries resources. As a result, Bangladesh and India have been in a race to exploit their shared waters for fisheries, often leading to competition on accessing the same resources. Bangladesh and India have also competed in water management [27]. Bangladesh has consistently argued for greater water sharing and equitable allocation of water resources, while India has opposed this, citing the need to protect its own interests.

Both countries have invested heavily in water management solutions, such as irrigation and flood control. In both countries pollution brought on by human activities is mentioned but not thoroughly covered in all aspects. Additionally, pollution or water scarcity are not portrayed as concerns for other nations [30]. To assess water quality, regular monitoring and evaluation of the water resources are required. Furthermore, integrated management of water resources including efficient water use, water conservation, and the protection of aquatic ecosystems should be established. To avoid the difficulty of comparing different water resources due to the varying characteristics used in the analysis, a mathematical method to calculate a water quality index (WQI) has recently been proposed to evaluate the quality of bodies of water [31]. Horton developed the first index in 1965, which had ten factors including dissolved oxygen (DO), pH, coliforms, electrical conductivity (EC), alkalinity, and chloride [32]. Several other water quality indices have been developed by various scientists and organizations for the assessment of water quality.

In order to develop a way to communicate water quality issues to scientists, decision-makers, and stakeholders, the Canadian Council of Ministers of the Environment (CCME) created the water quality index (WQI) that are now widely used [33]. Similarly, the National Sanitation Foundation (NSF) WQI also have both been utilized extensively around the world [34]. The CCME WQI has several benefits over other techniques, including adherence to various legal mandates and water uses, eligibility for water quality assessment in particular locations, flexibility in the selection criteria, and acceptance of missing data [32,35,36]. From a recent study it is revealed that water pollution is one of the major problems in this river [23]. To the best of our knowledge, there is no published data to date for evaluating the spatiotemporal changes in the Shari-Goyain River's water quality, assessing the suitability of the river's water, or identifying potential sources of water contamination, all of which are critical for safeguarding the river's ecological status. In order to achieve sustainable benefits from this river, it is necessary to evaluate the current situation and remain aware of what is occurring in various ecosystems throughout the river. The local people are sensitive to ongoing ecological change over time [37]; thus, the use of indigenous ecological information from local people can be an appropriate approach for participatory and adaptive conservation resolutions of natural resources. For this reason, scientists are now giving importance to the involvement of local people in the development and conservation of the environment and biodiversity. Therefore, the current study was conducted to evaluate the seasonal and spatial fluctuations of the river water's physicochemical parameters, to determine the variables and sources driving these variations by using multivariate statistical methodologies and CCME WQI, and thus finally identify the main cause of water pollution in the Shari-Goyain River according to local people's perceptions. Notably, this work is the first scientific assessment of coal mining drainage-mediated water pollution in the Shari-Goyain River. The findings of the current study are expected to provide new insights into the extent and sources of water pollution in the Shari-Goyain River and thus to reinforce management decisions for the protection and restoration of the river's water quality.

## 2. Materials and Methods

The flowchart in Figure 1 provides an overview of the research methodology used in the study.

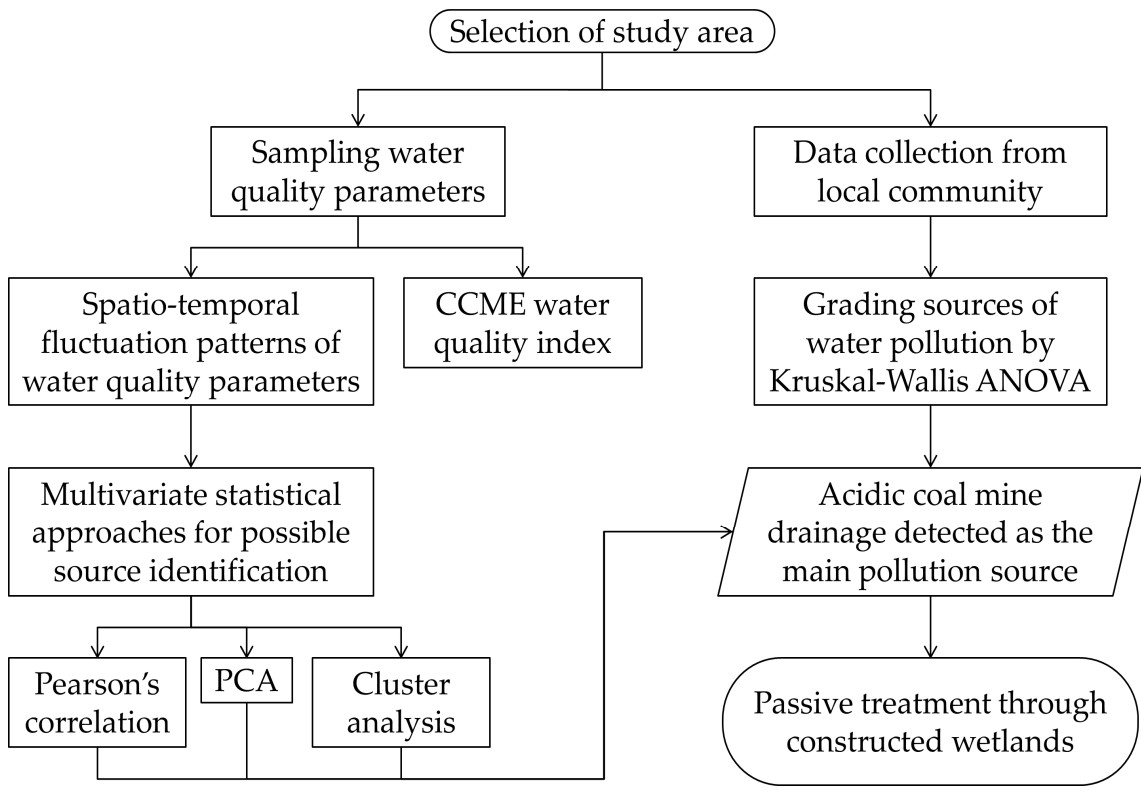

**Figure 1.** Flowchart of the research methodology.

### 2.1. Selection of Study Area

One of the main rivers in the Jaintia Hills district is the Myntdu River of Meghalaya. Mihmyntdu, which is not far from the town of Jowai, is where the Myntdu River begins. Its starting point is about 1420 m above the sea level. Umshariang, Lamu, and Lynriang Rivers are its main tributaries. The two tributaries, Umshariang, which enters from the west, and Lamu, which enters from the east, combine with the Myntdu River forming the tri-junction of Leshka, which consists of three rivers. The Myntdu-Leshka Dam was built transversely on this river. This river flows across the town of Jowai in the direction of Leshka, from there to Borghat village and then gradually advances towards Bangladesh in its course, and acquires the name Shari as it enters Bangladesh. The river is often described as the guardian and protector of the inhabitants of the Jowai region. It is considered a blessing for the people of the towns and villages that are nourished by its water [38]. The water of this river is polluted day by day, and the main causes of water pollution include runoff from agricultural lands, waste from residential activities, mining for coal and limestone, and mining for other minerals [39]. As a result, the pH level of this river is low which poses a serious threat to the waters and aquatic life [38]. Certain areas of the Myntdu River in the West Jaintia Hills area have changed their color to a bright sky blue demonstrating a very high acid content. In a report in 2012, the Meghalaya State Contamination Control Board (MSPCB) identified acid effluents from coal mines as likely the primary source of water pollution in this region [40]. Meghalaya is currently home to a number of large mining projects, including open-pit hard rock mines and strip mines for coal production. However, while hard rock mines usually only pose dust problems, coal mines emit polluted water, soil, and occasionally sludge. Thus, the Meghalaya's water supply in Jaintia Hills district is negatively impacted by these coal mining activities [41]. The Myntdu River receives polluted mine drainage, which finally flows to the Shari-Goyain River in Bangladesh. In this river numerous cases of massive fish deaths have been reported [18].

This study was conducted from December 2018 to November 2019 in the Shari-Goyain River. Six distant study sites were selected, covering both upstream and downstream of the

river *viz.*, Lalakhal ($S_1$), Sharighat ($S_2$), Mukhtola ($S_3$), Gowainghat ($S_4$), Jalurmukh ($S_5$), and Salutikar ($S_6$) (Figure 2). This river is important for its rich biodiversity; however, its fish production is continuously decreasing due to natural calamities, water pollution, and other problems [17,23]. From its point of arrival into Bangladesh and its confluence with the Surma River, there is a total of about 80 km [42]. The average daily recorded water flow rate at the $S_2$ site is about 126 m$^3$/s, which represents about 5200 mm annual runoff rate. This is a startlingly high flow rate, yet the catchment contains certain regions with exceptionally high rainfall that, at their highest, can surpass the average annual rainfall of 9000 mm. The average flow rate from November to March is 12 m$^3$/s; however, the dry season flows are substantially lower, with the average flow dropping to just 6 m$^3$/s in February, the driest month. During the wet months of May through August, the average flow increases to 260 m$^3$/s [42].

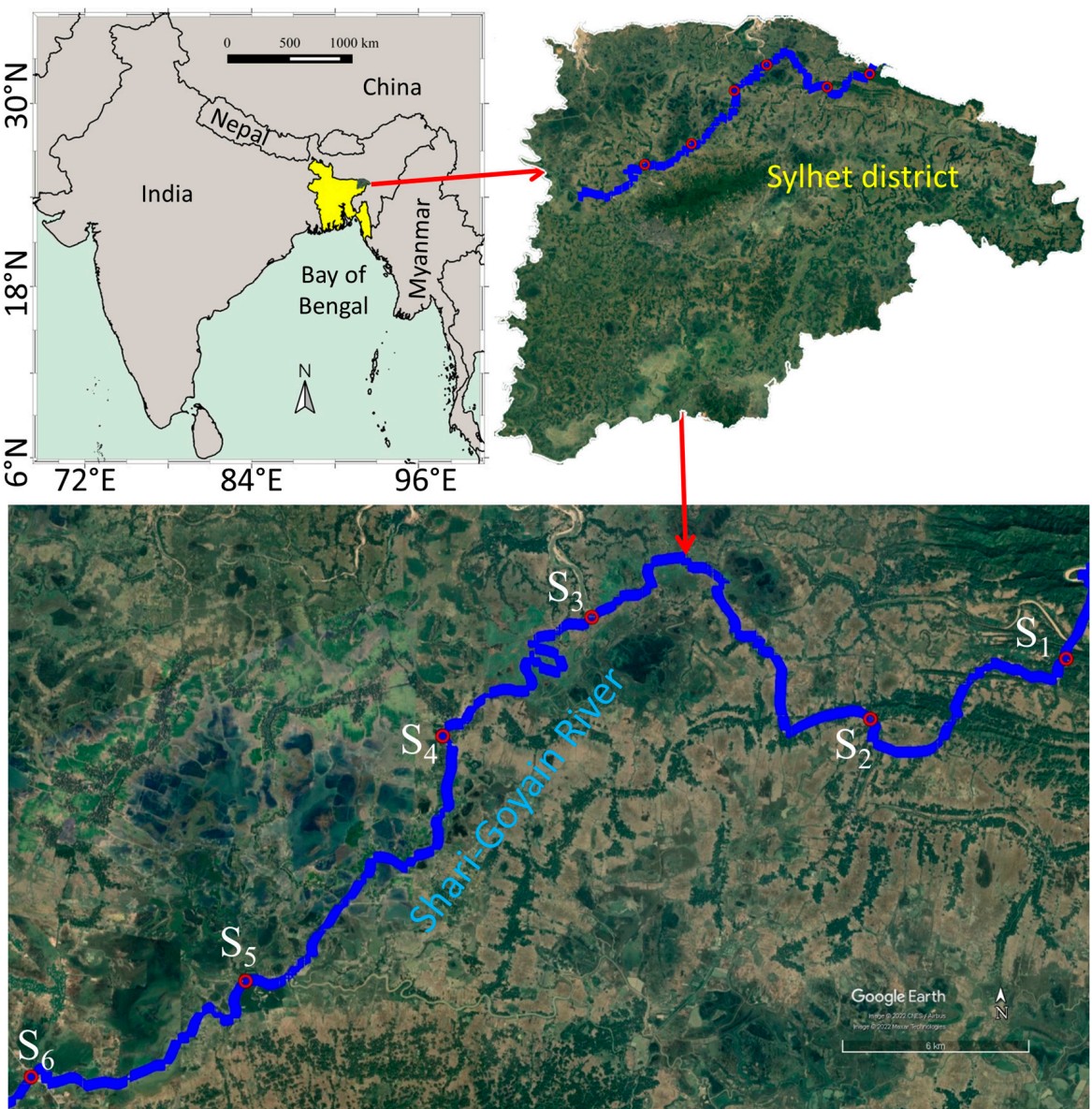

**Figure 2.** Inset map of Bangladesh showing the location of the Shari-Goyain River, and study sites in the Shari-Goyain River: Lalakhal ($S_1$), Sharighat ($S_2$), Mukhtola ($S_3$), Gowainghat ($S_4$), Jalurmukh ($S_5$), and Salutikar ($S_6$).

### 2.2. Water Sampling and Measurement

A digital multi-sensor (YSI Multi-Sensor, model: Professional Plus, brand: YSI, origin: USA) was utilized once a month to measure in situ physico-chemical variables such as pH, EC, total dissolved solids (TDS), DO, and the temperature of the water in six study sites with three replications. Additionally, a Secchi disk was used to measure the water transparency. These samplings were done during 9:00 to 11:00 a.m. according to local time of Bangladesh.

### 2.3. Calculation of CCME Water Quality Index

The CCME WQI model was developed from the British Colombia Water Quality Index (BCWQI) by the Water Quality Guidelines Task Group of the CCME in 2001 and has been used for evaluating the quality of surface water [31,43–45]. A committee created within the CCME developed the WQI as a standardized approach [33,46,47] which can be used for the sake of safeguarding aquatic life and evaluating the quality of the drinking water by following the standards, as we did in Bangladesh [44,48,49]. As it is not a standardized approach in Bangladesh, it was adopted following the methodology of Uddin et al. [44], Serajuddin et al. [48], and Hasan et al. [49]. From one station to another, the parameters for various measurements and sample techniques may vary. For at least four parameters, at least four samples are necessary [50,51]. The mathematical calculation of CCME WQI has been described below.

$$CCME - WQI = 100 - \left( \frac{\sqrt{F_1^2 + F_2^2 + F_3^2)}}{1.732} \right)$$

where $F_1$ reflects the number of variables (failed variables) whose objectives are not fulfilled, $F_2$ is the proportion of individual tests (failed tests) whose objectives are not met, and $F_3$ is the margin by which failed test values do not achieve their targets. After scaling the normalized sum of the excursions from objectives to produce a range between 0 and 100, an asymptotic function calculates $F_3$. When an individual focus exceeds (or falls short of, if the target is minimum) the objective, this is known as an "excursion" and is stated as follows [33,46,52].

$$F_1 = \left( \frac{Number\ of\ failed\ variables}{Total\ number\ of\ variables} \right) \times 100$$

$$F_2 = \left( \frac{Number\ of\ failed\ tests}{Total\ number\ of\ tests} \right) \times 100$$

When a test value is less than the objective value, the excursion is calculated as follows:

$$Excursion_i = \left( \frac{Failed\ test\ value_i}{Objective_j} \right) - 1$$

On the other hand, if the test value is more than the objective value, the excursion value is determined as follows:

$$Excursion_i = \left( \frac{Objective_j}{Failed\ test\ value_i} \right) - 1 \quad NSE = \frac{\sum_{n=i}^{n} Excursion_i}{Number\ of\ test} \quad F_3 = \left( \frac{NSE}{0.01nse + 0.01} \right)$$

The computations yielded the following rankings for the CCME WQI: bad (0–44), marginal (45–64), fair (65–79), good (80–94), and excellent (95–100) (Table 1) [33]. Numerous studies have made water quality assessments based on the legal requirements of their particular countries due to the CCME WQI's success [32,35,44,53,54].

**Table 1.** Rankings of CCME WQI values [33].

| Status | CCME WQI Value | Description |
|--------|----------------|-------------|
| **Poor** | 0–44 | The measurements frequently go far beyond the recommended levels for water quality. Aquatic life is in danger, degraded, or even extinct. |
| **Marginal** | 45–64 | The measurements frequently go above water quality standards by a wide margin. Aquatic life is frequently in danger or suffering. |
| **Fair** | 65–79 | The measurements occasionally and possibly significantly exceed water quality standards. Although aquatic life is protected, it may occasionally be in danger or compromised. |
| **Good** | 80–94 | The measurements almost never, and almost always by a small margin, exceed water quality standards. The safety and conservation of aquatic life are only slightly threatened or harmed. |
| **Excellent** | 95–100 | The measurements never, or very seldom, go above the recommended levels. There is no threat to or detriment to aquatic life. |

*2.4. Questionnaire Design and Data Collection*

Data (both qualitative and quantitative) regarding the local people's perceptions about the sources of water pollution in the Shari-Goyain River were collected from fishers, fish traders, boatmen, businessmen, local leaders, and riverbank communities through focus group discussions (FGDs), key informant interviews (KIIs), and personal interviews (PI). First, six FGDs, each with 8–12 members and 12 KIIs, were performed at the study sites. Based on the results of the FGDs and KIIs, a total of 11 sources of water pollution were primarily noted and a questionnaire was developed to rank the pollution sources. In developing the questionnaire, many questions concerning the causes of water pollution in the Shari-Goyain River were considered. The questionnaire had two sections regarding the demographic information of the respondents, and the causes of water pollution. The respondents were asked to respond based on an 11-point scale, scaling the highest- to lowest-ranked impacts on the Shari-Goyain River ecosystem. Based on the finalized questionnaire, a total of 74 people were personally interviewed. People were asked to rank the possible sources of water quality degradation and to give the respondents flexibility when answering the questions.

*2.5. Data Management and Analysis*

Considering the four seasons, namely, pre-monsoon (March–May), monsoon (June–August), post-monsoon (September–November), and winter (December–February), data were analyzed seasonally. Water quality attributes were calculated through mean values and standard deviation (SD) by using the Microsoft Excel. To identify possible relationships among the six study sites and four seasons one-way analysis of variance (ANOVA), PCA, and correlations were carried out by using the Statistical Package for Social Sciences (SPSS, version 20). PCA was used on the collected data to identify fundamental interrelationships amongst the parameters and to extort information about the correlation among variables that were analyzed in the water samples [55,56]. Pearson's correlation analysis was used for the assessment of associations among the water quality parameters of the sampling sites. A correlation coefficient value very near to 0 means that there was no linear relationship between variables, and if the value is close to $-1$ and 1, it means that there was a strong negative and positive relationship, correspondingly, between two variables for PCA [57]. However, the value of $r < 0.50$ with larger samples may have been highly statistically significant at $p < 0.01$, which meant that a low strength of correlation was noted. Where Kaiser–Meyer–Olkin (KMO) index and Bartlett's sphericity tests were used, we were able to factorize, check the normality, and efficiently justify the data. Values of correlation between variables and those of the partial correlations were compared by KMO index. When the

KMO index is close to 1, the PCA of the variables is suitable; but PCA is not applicable when the KMO index is close to 0. Usually, for pleasing analysis, this KMO index should be >0.5; thus, the KMO index for our study was satisfactory, as the value was 0.551 and the null hypothesis was rejected ($p < 0.05$). Z-scale standardization with mean and variance of zero and one was used, to minimize the variation in the variance of variables and to regulate the inequality in the variable sizes and the measurement units [58,59]. Estimation was done based on the correlation matrix of measured parameters, and the scores of PCA were achieved from the standardized variables data. To maximize the variance of the squared loadings, the varimax rotation was used. The number of factors was determined based on the Kaiser's condition [60]. Cluster analysis (CA) was used on the normalized data based on squared Euclidean distances as a measure of similarity for investigating similarity and dissimilarity in composition among sampling stations and seasons [61–63]. The proximity between two clusters is the increase in the squared error determined by Ward's method which is the most common method for classifying more perfect groups. For testing every parameter, distributions were centered and decreased before clustering. The clusters and their closeness with a deduction in the dimensionality of the main data are illustrated in a dendrogram [64]. Finally, Kruskal-Wallis ANOVA by ranks was employed to assess public perceptions about the major factors contributing to the deterioration of the Shari-Goyain River's water quality [26].

## 3. Results

### 3.1. Temporal and Spatial Variations in the Hydro-Chemical Parameters of River Water

The physicochemical parameters of the Shari-Goyain River water are found to fluctuate greatly and, thus, often exceed or fall below the suitable limits for the survival of aquatic flora and fauna (Tables 2 and 3). The mean value of water temperature was significantly lowest ($p < 0.05$) in the winter (20.74 ± 1.81 °C) and the highest in the monsoon season (28.93 ± 1.49 °C) (Table 2). Seasonally, the mean value of DO was found to fluctuate from the minimum value of 5.25 ± 0.77 mg/L in monsoon season to the maximum value of 7.22 ± 0.94 mg/L in winter. Spatially, lower DO values were observed at stations downstream of the river ($S_4$, $S_5$, and $S_6$). Temporally, the mean EC of water was significantly higher ($p < 0.05$) in pre-monsoon and winter seasons than in monsoon and post-monsoon seasons. Spatially, EC in $S_1$ was significantly higher than in other sites, but no significant difference was found between $S_1$ and $S_2$. Downstream, water EC shows a significant ($p < 0.05$) reduction, whereas no significant difference was found in EC for $S_4$, $S_5$, and $S_6$. The average value of TDS was found to fluctuate from the minimum of 32.46 ± 7.95 ppm in monsoon season to the maximum of 52.34 ± 11.38 ppm in winter. Water transparency ranged from 65.11 ± 44.61 cm in post-monsoon season to 75.97 ± 58.87 cm in pre-monsoon season and was significantly higher in the upstream sites ($S_1$ and $S_2$). The pH values in the study sites varied a lot (3.87–7.70); the average pH value was acidic during the whole year, but was highly acidic during the pre-monsoon season (3.87–6.61). Spatially, significantly lower pH values were recorded at two sites, $S_1$ and $S_2$, upstream of the river which was very close to the CMD discharge points upstream of the same river, the Meghalaya in India.

### 3.2. CCME WQI

The CCME WQI value of 33.40 at site 1 and 39.31 at site 2 indicated "poor" water quality, while the CCME WQI value of 48.63, 47.28, 51.30, and 45.91 at sites 3, 4, 5, and 6, respectively, indicated "marginal" water quality (Table 4). In this study, the river water quality indicates that the river is affected by water pollution. Due to the wastewater discharged during coal mining, the water quality of the two upstream sites is extremely contaminated and presents high danger for aquatic life. Calculations revealed that several anthropogenic pollutant sources may be similarly responsible for water pollution in other monitoring sites.

**Table 2.** Temporal fluctuation of water quality parameters in the Shari-Goyain River.

| Parameters | Seasons | | | | | | | |
|---|---|---|---|---|---|---|---|---|
| | **Winter** | | **Pre-Monsoon** | | **Monsoon** | | **Post-Monsoon** | |
| | **Range** | **Mean ± SD** | **Range** | **Mean ± SD** | **Range** | **Mean ± SD** | **Range** | **Mean ± SD** |
| Temp. (°C) | 17.90–26.20 | 20.74 ± 1.81 [c] | 24.20–29.90 | 27.46 ± 1.20 [b] | 25.70–32.50 | 28.93 ± 1.49 [a] | 23.60–31.80 | 27.26 ± 2.00 [b] |
| DO (mg/L) | 4.38–8.56 | 7.22 ± 0.94 [a] | 3.53–7.72 | 5.96 ± 1.03 [b] | 3.14–7.08 | 5.25 ± 0.77 [c] | 3.77–7.15 | 5.67 ± 0.89 [b,c] |
| EC (µS/cm) | 42.3–137.4 | 94.1 ± 21.3 [a] | 46.2–154.0 | 94.1 ± 31.5 [a] | 42.3–119.7 | 67.1 ± 19.0 [b] | 49.6–92.4 | 68.6 ± 12.3 [b] |
| TDS (ppm) | 29–76 | 52 ± 11 [a] | 22–76 | 45 ± 16 [b] | 23–59 | 33 ± 8 [c] | 23–47 | 34 ± 7 [c] |
| Trans. (cm) | 22–165 | 69.65 ± 37.97 [a,b] | 25–250 | 75.97 ± 58.87 [a] | 30.00–245.00 | 65.55 ± 43.63 [c] | 30–203 | 65 ± 44.50 [b,c] |
| pH | 4.28–7.70 | 6.01 ± 0.90 | 3.87–6.61 | 5.76 ± 0.91 | 5.07–6.96 | 6.50 ± 0.29 | 5.28–6.81 | 6.24 ± 0.38 |

Values for each variable with a different superscript letter indicate statistically significant differences at $p < 0.05$.

**Table 3.** Spatial fluctuation of water quality parameters in the Shari-Goyain River.

| Sites | Water Quality Parameters (Mean ± SD) | | | | | |
|---|---|---|---|---|---|---|
| | Temperature (°C) | DO (mg/L) | EC (µS/cm) | TDS (ppm) | Transparency (cm) | pH |
| $S_1$ | 24.63 ± 3.19 | 6.27 ± 0.98 | 104 ± 27 [a] | 53.44 ± 14.53 [a] | 144.98 ± 45.99 [a] | 5.44 ± 0.89 [a] |
| $S_2$ | 25.96 ± 3.88 | 6.24 ± 1.19 | 97 ± 26 [a,b] | 48.19 ± 13.68 [a,b] | 88.03 ± 49.00 [a,b] | 5.72 ± 0.85 [b] |
| $S_3$ | 25.89 ± 3.48 | 6.38 ± 1.16 | 82 ± 23 [b,c] | 40.38 ± 12.02 [b,c] | 45.96 ± 11.05 [b] | 6.04 ± 0.55 [c] |
| $S_4$ | 26.45 ± 3.38 | 5.85 ± 1.10 | 73 ± 15 [c,d] | 37.41 ± 9.77 [c,d,e] | 45.95 ± 9.82 [c] | 6.45 ± 0.28 [c] |
| $S_5$ | 26.91 ± 3.40 | 5.69 ± 1.24 | 64 ± 14 [d] | 32.13 ± 7.93 [d,e] | 44.33 ± 7.27 [c] | 6.55 ± 0.28 [c] |
| $S_6$ | 26.75 ± 3.80 | 5.72 ± 1.21 | 66 ± 18 [d] | 33.43 ± 10.01 [c,d,e] | 45.18 ± 6.78 [c] | 6.57 ± 0.39 [c] |
| Reference value | 20–30 [65] | 4–6 [66] | 100–2000 [67] | <600 [68] | 30–40 [69] | 6.5–8.5 [65,70] |

Values for each variable with a different superscript letter reflect statistical variations across sites at $p < 0.05$.

**Table 4.** Water quality index results.

| Study Site | CCME WQI Score | Water Quality Status |
|---|---|---|
| Site 1 | 33.40 | Poor |
| Site 2 | 39.31 | Poor |
| Site 3 | 48.63 | Marginal |
| Site 4 | 47.28 | Marginal |
| Site 5 | 51.30 | Marginal |
| Site 6 | 45.91 | Marginal |

*3.3. Relations among Water Quality Parameters and Possible Source Identification*

Based on Pearson's correlation coefficients (r), Table 5 displays the correlation matrix of the examined water quality parameters. A highly significant positive correlation (at 0.01 significance level) was found between EC-TDS (r = 0.945), DO-TDS (r = 0.544), EC-transparency (r = 0.514), transparency-TDS (r = 0.493), DO-EC (r = 0.422), and pH-temperature (r = 0.246). Furthermore, a significant positive relationship was observed between DO and transparency at the significance level of 0.05. The findings made it quite evident that there was a highly significant (correlation is significant at the 0.01 level) negative correlation between temperature-DO (r = –0.756), pH-EC (r = –0.607), pH-TDS (r = –0.599), temperature-TDS (r = –0.560), pH-transparency (r = –0.538), pH-DO (r = –0.414), and temperature-EC (r = –0.391). Additionally, temperature and transparency showed a significant positive linear relationship at a significance level of 0.05. The very strong and significant correlations showed that the parameters had come from similar sources.

**Table 5.** Correlation matrix of water quality parameters based on Pearson's correlation coefficients.

| | Temperature | DO | EC | TDS | pH | Transparency |
|---|---|---|---|---|---|---|
| Temperature | 1 | | | | | |
| DO | –0.75 ** | 1 | | | | |
| EC | –0.39 ** | 0.42 ** | 1 | | | |
| TDS | –0.56 ** | 0.54 ** | 0.94 ** | 1 | | |
| pH | 0.24 ** | –0.41 ** | –0.60 ** | –0.59 ** | 1 | |
| Transparency | –0.15 * | 0.14 * | 0.51 ** | 0.49 ** | –0.53 ** | 1 |

** Correlation is significant at the 0.01 level (2-tailed). * Correlation is significant at the 0.05 level (two-tailed).

To investigate the underlying correlations between the water quality variables of all monitoring sites as well as to determine their characteristics, a PCA based on the correlation matrix was performed. The scree plot (Figure 3) reveals the sorted Eigen values as a function of the PC number in descending order. After the second Eigen value, the slope changes dramatically and two components are retained (Figure 3). Therefore, the PCA of the total dataset retained two principal components (factors 1 and 2) based on the Eigen values greater than one, which explained about 79.17% of the total variance in the studied water

quality parameters (Table 6). The first factor ($PC_1$) contributed 58.46% of the total variance because of strong positive loadings of pH, EC, and TDS, and a strong negative loading of transparency. This phenomenon can be attributed to the presence of acidic runoff and contaminants from various sources. The second factor ($PC_2$) explains 20.71% of the total variance. It had a strong positive loading on DO, a strong negative loading on temperature, a moderate positive loading on TDS, and a poor positive loading on EC, which could be attributable to natural and climatic drivers of the river water. Overall, these PCA analyses revealed the potential sources of water contamination in the Shari-Goyain River water. This contamination is the result of both natural and manmade sources.

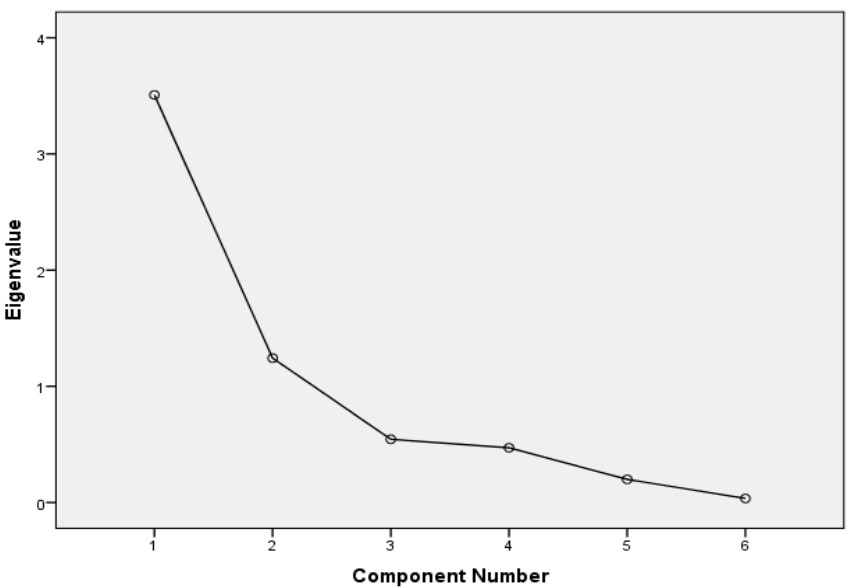

**Figure 3.** Scree plot of the loaded variables.

**Table 6.** Loadings of experimental variables by principal component analysis for the whole dataset.

| | | Factors | |
|---|---|---|---|
| Sl. So. | Variables | Factor 1 ($PC_1$) Anthropogenic Acidic Factor | Factor 2 ($PC_2$) Natural Factor |
| 01 | Temperature (°C) | –0.126 | –0.916 |
| 02 | DO (mg/L) | 0.191 | 0.898 |
| 03 | EC (μS/cm) | 0.825 | 0.374 |
| 04 | TDS (ppm) | 0.765 | 0.534 |
| 05 | Transparency (cm) | –0.789 | –0.208 |
| 06 | pH | 0.831 | –0.077 |
| | Eigen value | 3.508 | 1.243 |
| | Percentage of variance explained | 58.46 | 20.71 |
| | Cumulative % variance | 58.46 | 79.17 |

### 3.4. Cluster Analysis Based on Spatial and Temporal Variations in River Water Quality

Spatial and temporal similarities and dissimilarities of water quality parameters were grouped using the CA method. Figure 4 displays the dendrograms of water quality parameters in seasons and stations, respectively, derived by Ward's method during the study period. Temporally, all four seasons formed two groups based on the CA results. The first cluster included two seasons (winter and pre-monsoon) and the second cluster also comprised another two seasons (monsoon and post-monsoon) (Figure 4A). Monsoon and post-monsoon seasons were grouped into a single cluster, which may be related to the effects of intense rainfall, which may have diluted the pollutants' concentrations and thoroughly distributed them along the river's downstream reaches. Spatially, the sample sites were

found to be clustered into two primary groups according to the CA results (Figure 4B), indicating that the water pollutants in each group came from comparable anthropogenic sources and that the sampling sites within each group have similar characteristics. Here, cluster one is composed of stations $S_1$ and $S_2$, which displayed similar quality attributes that correspond to higher pollution loads compared to the other cluster. Stations $S_1$ and $S_2$ were located upstream of the Shari-Goyain River in the Bangladesh section, whereas $S_1$ was situated downstream of the coal-contaminated wastewater discharge points of the Myntdu River in Meghalaya, India. Cluster two is composed of stations $S_3$, $S_4$, $S_5$, and $S_6$. Cluster two corresponds to moderate contaminations as upstream pollutants were diluted with the water from the Piyain, the Kapna, and their tributaries.

### 3.5. People's Perceptions about the Sources of Water Pollution

The main sources of water pollution were underscored by the respondents (Table 7). Kruskal-Wallis one-way ANOVA by ranks was used to accumulate the respondents' perceptions about the major sources of water pollutants in the Shari-Goyain River. People were asked to rank the possible sources of water quality degradation in a numbering system ranging from 1 to 11, demonstrating the highest- to lowest-graded impacts on the Shari-Goyain River ecosystem, respectively. According to the people's perception, the main cause of water pollution in the river was coal mine drainage, followed by industrial run-off, sewage from households, agricultural run-off, sewage from markets, poisoning from fishing, blast fishing using explosives upstream, stone crushing waste, navigation, tourism, and sand mining (Table 7). Notably, there was a statistically significant difference between each of the sources that the respondents recognized and ranked. According to the local people's perception, the Shari-Goyain River receives a high load of CMD 2–3 times a year in the pre-monsoon season, especially during April and May, and it receives a lower amount of effluents throughout the rest of the year. During that period, mass mortalities of fish and other aquatic organisms were observed in the river. Highly acidic pH might be the main factor there. In the pre-monsoon season, pH is reduced up to 3.87 (Table 2). Although mortality was mainly found in pre-monsoon season, the residents of S1 and S2 explained that mortality was also observed in December (winter) and is associated with sudden rainfall.

### 3.6. Mitigation of the Transboundary Water Management Problem: Passive Treatment for Acid Mine Drainage as a Nature-Based Solution

An outline of potential passive treatment approaches has been constructed based on the previous literature (Figure 5). Constructed wetlands, vertical flow wetlands, and bioreactors are a few examples of biological passive treatment methods that mostly rely on bacterial activity and may incorporate organic matter to promote microbial sulfate reduction and adsorb pollutants. Limestone and other alkalinity-producing minerals are brought into direct contact with AMD (direct treatment), or with freshwater that is a step upstream of the AMD, using geochemical systems. The chemistry and circumstances of AMD must be considered before choosing an appropriate treatment method [71]. Results from several passive treatment approaches were combined into a thorough USBM (U.S. Bureau of Mines) publication [72], which illustrated a design decision tree that classified polluted mine waters into chemical classes based primarily on alkalinity and acidity, and secondarily on metal contaminants [73]. The design decision tree also identified the passive treatment technologies that were most suitable for the specific water chemistry conditions (Figure 5). This distinction allowed later researchers and designers to better concentrate on important geochemical demands by explaining a large portion of the varied performance of current systems. Numerous scholars have since adopted and changed the design decision tree [71,73,74].

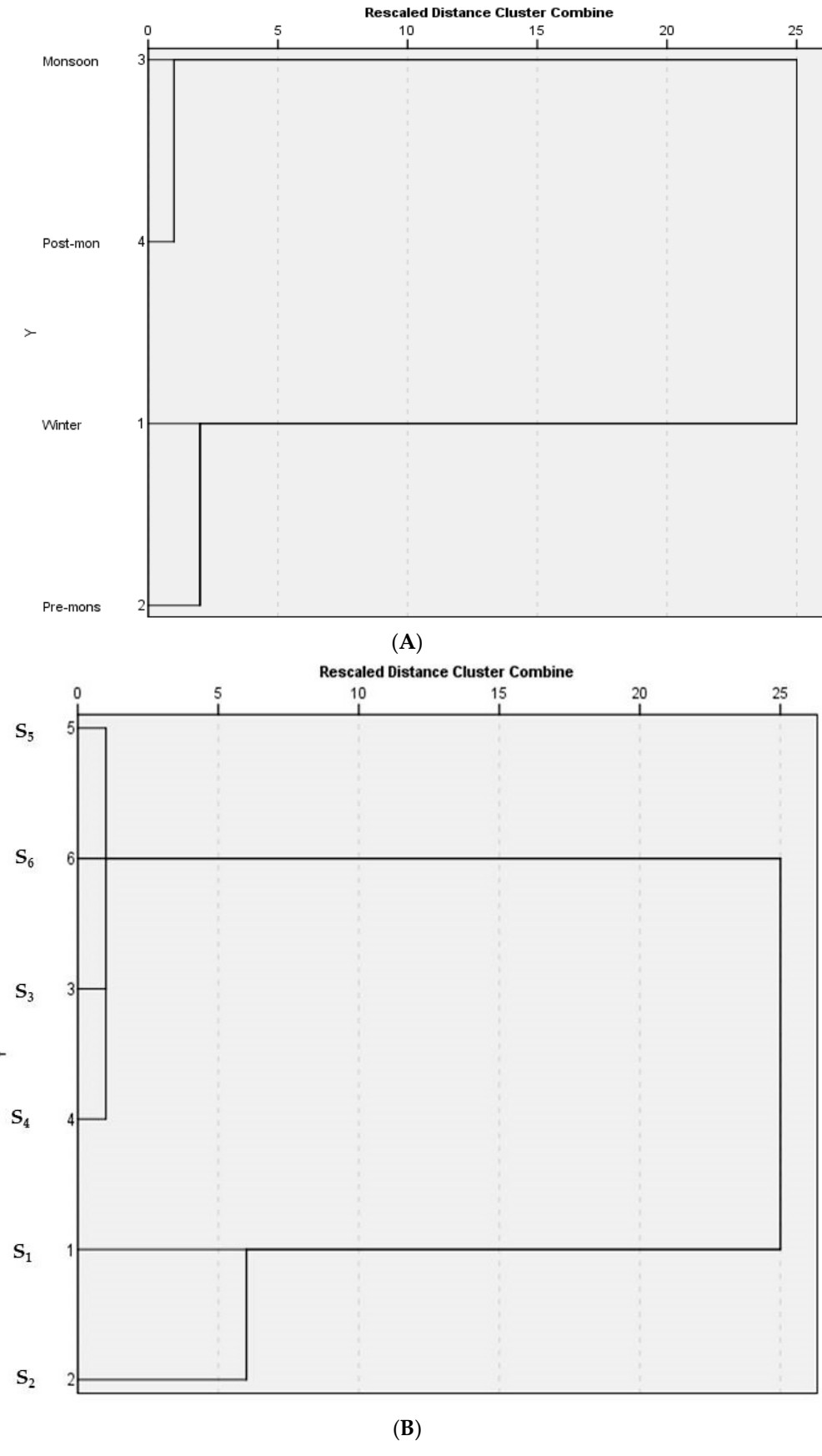

**Figure 4.** Dendrogram based on hierarchical clustering (wards method) for seasons (**A**) and for six sites (**B**).

**Table 7.** Results of Kruskal-Wallis test for ranking the sources of water pollution.

| Sl. No. | Sources of Water Pollution | Mean Rank | Position |
|---|---|---|---|
| 1 | Coal mine drainage | 37.5 | A |
| 2 | Industrial run-off | 177.5 | B |
| 3 | Sewage from households | 233.5 | C |
| 4 | Agricultural run-off | 318.5 | D |
| 5 | Sewage from markets | 408.5 | E |
| 6 | Poisoning from fishing | 469.5 | F |
| 7 | Blast fishing using explosives | 516.5 | G |
| 8 | Stone crushing waste | 535.5 | H |
| 9 | Navigation | 582.5 | I |
| 10 | Tourism | 588.5 | J |
| 11 | Sand mining | 614.5 | K |
| | Chi-square | 494.449 | |
| | Degree of freedom | 10 | |
| | Asymptotic significance | 0.000 | |

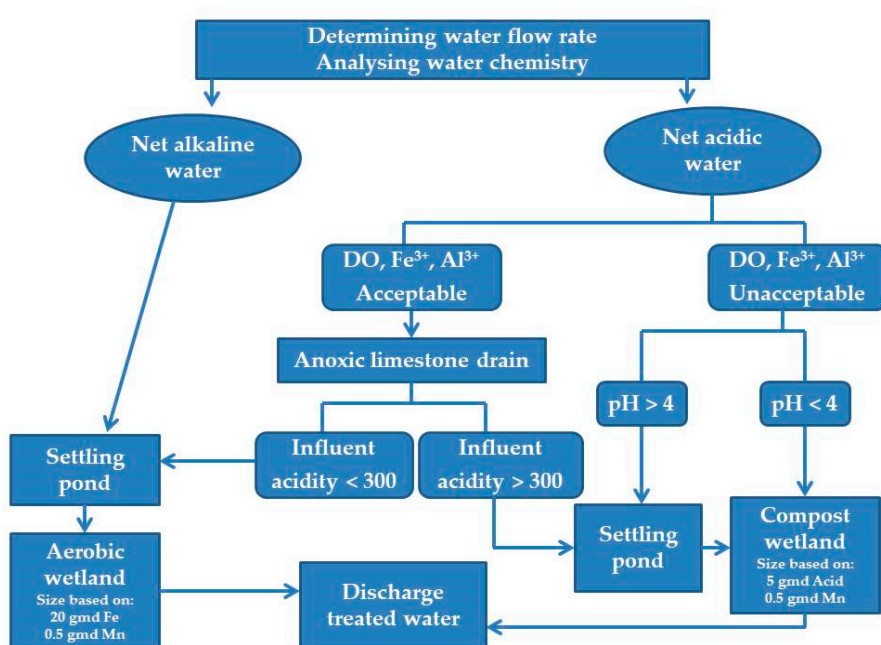

**Figure 5.** A primary decision tree for selecting a passive CMD treatment system, modified from Kleinmann et al. [73].

## 4. Discussion

A statistically significant variation ($p < 0.05$) was found in the physicochemical parameters among the different study sites and seasons. The water temperature causes substantial environmental effects by influencing the chemical, physical, and biological parameters[75]. It appeared that the mean temperature in all sampling sites and seasons was within the acceptable limit (20–30 °C) according to Bangladesh's Environmental Conservation Rules [76]. In the Shitalakhya River, average temperature values throughout the pre-monsoon, monsoon, and post-monsoon seasons were 28.00–28.15 °C, 30.30–30.65 °C, and 24.80–24.97 °C, respectively, which are relatively similar with the present study [77]. Kumari et al. [78] found that the solubility of ambient DO in river water decreases with the increase in water temperature, which supports the present study. The concentration of DO is a critical factor in keeping aquatic habitats in balance. It is a primary factor in determining the quality of water supplies. The river's flowing water has a concentration of roughly 10 mg/L [79]. However, the findings of the present investigation reveal that the DO readings were within the permitted level ($\geq$5 mg/L) for fisheries' production [76].

According to a comparable guideline for riverine fishes, warm-water adult fishes should be protected at DO concentrations above 5.5 mg/L, or above 6 mg/L for other life stages [80]. Discolored, turbid, or exceptionally clear water indicates CMD pollution [81]. Because there are so many iron hydroxide particles floating, water can sometimes have a distinctive yellowish-reddish-brown color. The turbidity of the CMD water generally decreases downstream as the iron and aluminum flocculate and salts precipitate with decreasing pH values. As a result, despite an unsuitable environment for aquatic organisms in acidic waters, it could also be exceptionally clear in cases of low suspended matters, and thus may give the wrong impression of being of good quality from visual observation [81]. This phenomenon was also observed in the Shari-Goyain River during coal mining flushes.

Generally, pH is the indicator of the acidic or alkaline status of water. The standard value for any purpose in terms of pH is 6.5–8.5 [76]. According to the U.S. Environmental Protection Agency (EPA) [82], the suitable range of pH is 6.5–9.0 for freshwater aquatic life. The most widely used indicator of CMD pollution is pH [83,84]. A possible reason for the high acidity in the Shari-Goyain River is the CMD that comes from upstream of the Shari-Goyain River. Gogoi & Saikia [84] studied water quality parameters concerning CMD at the middle reaches of the Dikhow River in Assam, and found a pH of $5.53 \pm 0.25$ from April to July when the river frequently comes into contact with the CMD [84]. Dave & Tipre [81] correlated CMD waters with low levels of pH (<5.5) which supports the present study. In Meghalaya, several devastating coal mining activities undertaken without environmental consideration have adversely affected the water resources of the Jaintia Hills district of Meghalaya, leading to streams with a very low pH (usually 3–5) [41]. Another study shows that the bodies of water in the mining areas of Meghalaya have been adversely affected by the contamination of acid mine drainage originating from coal mining operations. Low pH ($\leq 4.0$), and low dissolved oxygen associated with coal would have resulted in the decline or complete loss of fish fauna in the wetlands of the coal mining areas [85]. However, polluted acid mine drainage finds its way into the Myntdu River and flows downstream to the Shari-Goyain River. A possible reason for the high acidity in the pre-monsoon season is the CMD that comes from upstream of the Shari-Goyain River. Thus, upstream polluted water has a direct impact on the fish and wildlife resources of vast areas of aquatic habitat in the Shari-Goyain River of Bangladesh.

The relationships between water quality measures in a river environment offers crucial information about potential sources and pathways of variables [77,86]. The very strong and significant correlations showed that the parameters had come from similar anthropogenic sources [77,86]. The Kaiser criterion [60] was used to assess how many principal components (PCs) should be kept. The PC loadings were divided into three categories: strong, moderate, and weak, with absolute loading values of >0.75, 0.75–0.50, and 0.50–0.30, respectively [59]. An Eigen value gives a measure of the significance of the factor, and the factors with the highest Eigen values are the most significant. Eigen values of 1 or greater are considered significant [87]. Due to their poor importance, variables that have Eigen values less than one were deleted [79]. The PCA of the total dataset retained two principal components (factors 1 and 2) based on Eigen values greater than one, thus explaining about 79.17% of the total variance in the studied water quality parameters (Table 6). The first factor (PC$_1$) contributed 58.46% of the total variance, because of strong positive loadings of pH, EC, and TDS, and a strong negative loading of transparency. This factor can be attributed to acidic drainage pollution sources where these variables originated mainly from CMD [84,88]. Ray & Dey [88] analyzed the CMD waters collected from northeastern Assam in India and found the CMD has a low pH and high TDS. The dissolution of oxidized pyritic minerals found in coal and waste rock is what leads to this extremely acidic environment. In the presence of pyrite, the rate of oxidation increases, causing sulfate ions to develop [88]. However, there is no substantial salinity indicated by the EC values. The second factor (PC$_2$) explains 20.71% of the total variance. It had strong positive loading on DO, a strong negative loading on temperature, a moderate positive loading on TDS, and a poor positive loading on EC, which could be attributable to natural and climatic drivers of

the river water. Overall, these PCA analyses identified the potential contamination sources of Shari-Goyain River water. This contamination is a result of mixed sources, including those of both natural and anthropogenic origins.

Temporally, the first cluster included two seasons (winter and pre-monsoon), and the second cluster also comprised two seasons (monsoon and post-monsoon). Monsoon and post-monsoon seasons were grouped into a single cluster, which may be related to the effects of intense rainfall, which may have diluted the pollutants' concentrations and thoroughly distributed them along the river's downstream reaches. Spatially, the sample sites were found to be clustered into two primary groups according to the CA results, indicating that the water pollutants in each group came from comparable anthropogenic sources and that the sampling sites within each group have similar characteristics. Here, cluster one is composed of stations $S_1$ and $S_2$, which displayed similar quality attributes that correspond to higher pollution loads compared to the other cluster. Stations $S_1$ and $S_2$ were located upstream of the Shari-Goyain River, whereas $S_1$ is situated downstream of the coal-contaminated wastewater discharge points of the Myntdu River in Meghalaya, India. Cluster two is composed of stations $S_3$, $S_4$, $S_5$, and $S_6$. Cluster two corresponds to moderate contaminations as upstream pollutants were diluted with the water from the Piyain, the Kapna, and their tributaries.

Scientists and policy specialists must study how local communities perceive environmental pollution and its effects [89]. According to the people's perception, the main cause of water pollution in the river was coal mine drainage, which causes mass mortality incidences in aquatic organisms. The community living around the Shari-Goyain River has observed that the waterway accumulates a large quantity of CMD during the pre-monsoon season, particularly in April and May, while a lesser amount of effluents are present during the remaining months of the year. During the pre-monsoon season, CMD causes the water to become rapidly polluted, leading to the deaths of a large number of fish and other aquatic life[18]. Near the mining site, coal releases large quantities of heavy metals [90,91]. In addition to altering cellular biochemistry, developmental deformities, DNA damage, and general chromosomal abnormalities, these metals are also extremely poisonous [91–93]. The detrimental effects of coal dust include modifications to benthic habitats, the removal or modification of microhabitats for reproduction, and changes to the biological and biogeochemical cycles of aquatic resources and their populations [91,94,95]. International researchers discovered numerous detrimental effects of CMD (in coal mining zones) in the gills of *Oncorhynchus mykiss*, as well as juvenile and adult mortality in *Pimephales promelas*, and a mutagenic effect in *Oryzias latipes* [91,96,97]. Eight research studies on the effects of coal on aquatic systems have been conducted in Colombia [91]. These reported that carboniferous sediments from ports are positively correlated with polycyclic aromatic hydrocarbon derivatives [91,98]. Shi & He [89] examined the perception of residents in some coal mining areas of Shaanxi Province in China, where the main causes of environmental pollution were assessed. According to the public perception of the major factors contributing to environmental pollution in mining regions, the majority of locals believe that the environmental pollution in study areas is mostly brought on by industries associated with coal and is made worse by weak law enforcement, inefficient management in the mining districts, and low levels of environmental awareness among the local populace [89]. Fish and other aquatic fauna of the Shari-Goyain River have been prone to injury and death, resulting in an alarming decrease in the gross fish production in the river. CMD pollution has destroyed the natural fish breeding grounds of the river, leading to the further detriment of the aquatic ecosystem. In addition, these toxic pollutants have entered the bodies of cattle and humans through drinking water and fish consumption, leading to adverse health effects such as mental disorders, weakness, headaches, abdominal cramps, diarrhea, and anemia. This has consequently increased medical costs, leading to a rapid decrease in the socio-economic status of the people that depend on the riverine resources. Therefore, it is essential to take proper steps to ensure the survival of the aquatic ecosystem and the livelihoods of thousands of riverbank dwellers, especially fishers. Both active and

passive methods can be used to treat acidic CMD [99]. However, passive treatment is the most suitable method for treating modest to moderate AMD discharges, but regular maintenance, inspection, and possibly renovation are usually needed [71,73,74]. From the above discussions, one potential low-cost option for mitigating the impacts of coal mine drainage as the main source of pollution might be nature-based solutions, particularly passive mine water treatment approaches through constructed wetlands [100]. Constructed wetlands combine naturally occurring biogeochemical, geochemical, and physical processes to remediate acid mine drainage [99], which was identified as the main threat to water quality and fauna in the transboundary Shari-Goyain River. For this purpose, typical local plants that can reduce the pollutant input into the water body using phytoremediation should be identified [13]. Moreover, this will support the catchment ecosystem services. In this way, the passive treatment system will support a low-cost solution for the transboundary water quality problem.

## 5. Conclusions

The Shari-Goyain River is one of the primary sources of water consumption for humans and cattle, domestic washing, agricultural irrigation, and industrial use which is being polluting daily. To assess the spatial and seasonal variations of surface water quality, CCME WQI and multivariate statistics including Pearson's correlation, PCA, and CA were used. The water quality of the river is far from desirable for aquatic life, and it is being impacted and deteriorated by external drivers. This demonstrated that the pollutants in the river catchment region coming from the upstream coal mining areas have an impact on the river's water. Finally, Kruskal-Wallis one-way ANOVA evaluated the perceptions about the main causes of water contamination in the Shari-Goyain River and found coal mine drainage to be the main pollutant source. Therefore, a few permanent water quality monitoring stations should be installed throughout the river, coupled with thorough ecological studies. Since the Shari-Goyain River crosses international borders, it is imperative to take immediate action to develop a solution for protecting the environment of this river by assembling a team of scholars and administrators from Bangladesh and India. A potential low-cost option for mitigating the impacts of coal mine drainage might be nature-based solutions, particularly passive mine water treatment approaches through constructed wetlands.

**Author Contributions:** Conceptualization, D.P., M.M.H., P.S. and M.K.; methodology, D.P., M.M.H., A.H.-A.-R., P.S. and M.K.; software, D.P., A.H.-A.-R. and B.S.; validation, D.P., M.M.H., A.H.-A.-R., M.A.H., P.S. and M.K.; formal analysis, D.P., A.H.-A.-R. and M.K.; investigation, D.P., M.M.H., P.S. and M.K.; resources, P.S. and M.K.; data curation, D.P. and A.H.-A.-R.; writing—original draft preparation, D.P. and M.K.; writing—review and editing, D.P., M.M.H., A.H.-A.-R., B.S., M.A.H., P.S. and M.K.; visualization, D.P., B.S. and A.H.-A.-R.; supervision, M.M.H. and M.K.; project administration, M.K.; funding acquisition, M.K. All authors have read and agreed to the published version of the manuscript.

**Funding:** The study was funded by the National Agricultural Technology Program: Phase II Project (Sub-Project ID #035) of the Bangladesh Agricultural Research Council (BARC), Dhaka, Bangladesh.

**Institutional Review Board Statement:** Not applicable.

**Informed Consent Statement:** Not applicable.

**Data Availability Statement:** Data will be available from corresponding authors based on reasonable request.

**Acknowledgments:** Debasish Pandit would like to thank the Bangabandhu Science and Technology Fellowship Trust, Ministry of Science and Technology of Bangladesh for the granting of his Ph.D. fellowship. The authors appreciate the fishing community and other riverbank stakeholders of the Shari-Goyain River. Also, the authors appreciate the editor and the anonymous reviewers of this journal for their insightful criticisms, comments, and ideas.

**Conflicts of Interest:** The authors declare no conflict of interest.

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
