# Peer review of "Spatiotemporal Variations in Water Quality of the Transboundary Shari-Goyain River, Bangladesh"

_sustainability, doi:10.3390/su15065218_

Round 1
Reviewer 1 Report
In page 2, line 83. it should be written as"Altogether, there are 54 transboundary rivers....."
In page 2, line 87. The sentence should be rewritten.
In page 3, line 104-105. The sentence "To assess the water quality of bodies of water,....." it should be rewritten. So also the next sentence please.
In page 3, line 121. The sentence should be written as "to the best of our knowledge, there is no any published data to date on evaluating the ......."
Author Response
Dear respected reviewers,
Please check the attachment for details responses to your constructive comments and suggestions.
Regards
Authors.

Reviewer 2 Report
This manuscript examines coal mining on water quality. The manuscript is attractive and practical.Paying attention to the following comments and suggestions will improve the manuscript.
1- Make the title of the research simpler.
2- In the abstract, objective results of statistical tests and their significance on water quality should be presented.
3- In the literature review, in the introduction section, newer references should be used.
4- At the end of the introduction, research innovation should be presented.
5- In the method section, briefly state the measurement method of each variable. Like pH, EC, ....
6-Line 221 , 222: E????????? and NSE
7-Provide a flowchart of the research methodology
8-It is necessary to discuss the reason of People's perceptions about the sources of water pollution in the discussion section.
Author Response
Dear respected reviewers,
Dear respected reviewers,
Please check the attachment for details responses to your constructive comments and suggestions.
Regards
Authors.

Reviewer 3 Report
Dear Authors,
I would like to thank you the opportunity to review the manuscript.
Here are some notes about the submitted manuscript:
- The manuscript presents interesting results, which are relatively well organized and systematized, but the novelty and practical applicability of this study should be highlighted more. Also, it will be useful to include some information regarding the economic impact of the work;
- The figures need to be redone. It is very difficult to understand what is described inside;
- The map of the study area should show the expanded area for the reader to understand which region of the globe is;
- Some citations are in disagreement with the norms of the journal;
- As it is a case study, I imagined that there would be more information about the physical and human characteristics of the area. You should explorate this information in your paper;
-A basic flowchart of the suggested methodology should be presented in the paper. Thus, the readers can easily follow the application procedures;
- The performance metrics part is very important for the evaluation of application results. For this aim, the performance metrics as NSE, RMSE, etc. from literature should be calculated. Then, they can be given in a table. Because of the importance of the assessment of results, these metrics are widely used in the literature, especially in the forecasting studies. The new and main papers are suggested below. They should be benefited and cited;
- Conclusions part can be improved in the paper. Here is presented in a general concept;
- Is the used methodology in the paper valid for all areas or is there any limitation or classification for the application?
I hope I have contributed to my comments and suggestions.
Bests regards.
Author Response

(The authors gave the same response as above.)

Round 2
Reviewer 2 Report
ِDear editor
The author has answered the questions and comments in a favorable way.
Manuscripts are acceptable.